# Effects of Orientation and Dispersion on Electrical Conductivity and Mechanical Properties of Carbon Nanotube/Polypropylene Composite

**DOI:** 10.3390/polym15102370

**Published:** 2023-05-19

**Authors:** Dashan Mi, Zhongguo Zhao, Haiqing Bai

**Affiliations:** 1School of Mechanical Engineering, Shaanxi University of Technology, Hanzhong 723001, China; 2School of Materials Science and Engineering, Shaanxi University of Technology, Hanzhong 723001, China; zhaozhongguo@snut.edu.cn

**Keywords:** shish-kebab structure, polypropylene, carbon nanotubes, mechanical properties, electrical conductivity

## Abstract

The orientation and dispersion of nanoparticles can greatly influence the conductivity and mechanical properties of nanocomposites. In this study, the Polypropylene/ Carbon Nanotubes (PP/CNTs) nanocomposites were produced using three different molding methods, i.e., compression molding (CM), conventional injection molding (IM), and interval injection molding (IntM). Various CNTs content and shear conditions give CNTs different dispersion and orientation states. Then, three electrical percolation thresholds (4 wt.% CM, 6 wt.% IM, and 9 wt.% IntM) were obtained by various CNTs dispersion and orientations. Agglomerate dispersion (Adis), agglomerate orientation (Aori), and molecular orientation (Mori) are used to quantify the CNTs dispersion and orientation degree. IntM uses high shear to break the agglomerates and promote the Aori, Mori, and Adis. Large Aori and Mori can create a path along the flow direction, which lead to an electrical anisotropy of nearly six orders of magnitude in the flow and transverse direction. On the other hand, when CM and IM samples already build the conductive network, IntM can triple the Adis and destroy the network. Moreover, mechanical properties are also been discussed, such as the increase in tensile strength with Aori and Mori but showing independence with Adis. This paper proves that the high dispersion of CNTs agglomerate goes against forming a conductivity network. At the same time, the increased orientation of CNTs causes the electric current to flow only in the orientation direction. It helps to prepare PP/CNTs nanocomposites on demand by understanding the influence of CNTs dispersion and orientation on mechanical and electrical properties.

## 1. Introduction

Many theoretical and experimental studies have been done on the effects nanofillers on the mechanical and electrical properties of nanocomposites in the last decade [1,2,3]. Applications using carbon nanotubes (CNTs) as nanofillers have increased [4,5,6,7], such as burning anti-dripping performance, electromagnetic shielding, and sensing element. Polymer blend-based multicomponent nanocomposites with superior mechanical and thermal properties due to the modification of phase morphology have been reported [8]. Although CNTs have good mechanical properties, only a small fraction of their conductivity and strength can translate into the matrix in which CNTs are embedded and twisted [9]. CNTs dispersion and orientation can affect the final performance of nanocomposites, but many related theoretical and experimental results are still controversial.

For example, some researchers show that well-dispersed CNTs are efficiently interconnected and thus decrease the electrical percolation threshold [10,11,12]. Tanabi et al. [13] disperse the CNTs via magnetization and calculate the dispersion state using microscopy images. Results showed that the electric conductivity could be increased significantly with the post-dispersing of CNTs. Burmistrov et al. [14] used two methods to disperse CNTs and carbon blacks. The twin-roller mixer can give a more uniform dispersion than the Haake Rheomix Polylab, resulting in a higher electrical conductivity.

Whereas other studies have shown the opposite effect, according to Mold et al. [15], small-size CNTs agglomerates make it easier to form conductive pathways and enhance the conductivity of PP/CNTs composites. Mei et al. [16] also suggested that shearing conditions in micro injection moldings can disperse and align the CNTs along the flow direction, which is unfavorable for constructing conductive pathways. Ferreira et al. [17] used agglomerate area ratios (A_R_) to calculate CNTs and found that the improved dispersion results in higher electrical resistivity. It is of great practical significance to clarify the effect of dispersion on conductivity in the contradictory experimental conclusions. For example, Tang [18] believed that a moderate CNTs dispersion can help to obtain excellent CNT-based composite sensors for structural health monitoring and flexible electronics.

In addition, aligned carbon nanotubes have been widely considered ideal nanofillers for manufacturing anisotropic polymer nanocomposites due to their unique nanostructure, high conductivity, and strength in the alignment direction. The presence of aligned CNTs can enhance the orientation of crystalline and polymeric chains along a particular axis, leading to anisotropic thermal phonon transport, electron conduction, and reinforcement properties [19]. On the other hand, compared with unoriented CNTs, the oriented nanofibers may get complex results in forming a conductive path. Zhou prepared PP/CNTs nanocomposites using microinjection molding and found that increased injection velocity can align CNTs, then increasing the probability of CNT-CNT contact, facilitating the enhancement of electrical conductivity in the flow direction [20]. In contrast, Monti et al. [21] tested the AC dielectric of injected PP/CNTs nanocomposites in the 10^1–^10^6^ HZ frequency range and proved that the lower shear stress leads to a rearrangement of the CNTs clusters, which is more efficient in electrical conduction.

The relationship between the CNTs distribution and properties plays an essential role in improving the conductivity of components, improving the mechanical strength, and preparing piezoresistive sensing applications. Although there are many studies on the orientation or dispersion of nanofillers, as far as we know, few of them have discussed the combined influence of orientation and dispersion. For this reason, this paper aims to uncover the impact of CNTs dispersion and orientation on mechanical and electrical conductivities. In the present work, PP was used as the matrix to prepare PP/CNTs nanocomposites, which has potential applications in the computer chip, electromagnetic shielding, and the dissipation of electrostatic discharge [3]. Compression molding (CM), conventional injection molding (IM), and interval injection molding (IntM) were used to provide different melt shear rates to change CNTs distribution. Then, the dispersion and orientation of CNTs were quantified and used to study the relationship with performance, which can help to prepare PP/CNTs nanocomposites on demanded mechanical and electrical properties.

## 2. Experimental Section

### 2.1. Materials and Sample Preparation

Isotactic polypropylene (iPP, trade name T30S) was purchased from Sinopec Maoming Petrochemical Co., Ltd., Maoming, China. CNTs were purchased from Suzhou Tanfeng Graphene Technology Co., Ltd., Suzhou, China. CNTs were multi-walled with an average diameter of 50 nm, an average length of 20 μm, and a middle surface area of 300 m^2^/g, according to the supplier, with a carbon purity of 90%.

Figure 1 shows a series of polymer composites with filler contents of 0, 1, 2, 4, 6, and 9 wt.% which were blended using a micro-conical double-screw extruder (SJZS-10A, Wuhan ruiming Co., Ltd., Wuhan, China) at a barrel temperature of 200 °C. Then, three molding methods were used to provide different shear stress, such as compression molding (CM), injection molding (IM), and interval injection molding (IntM). All samples were labeled according to CNTs content and molding method. For example, 2IntM means that the CNTs content was 2 wt.% and was molded using IntM. The central processing conditions were denoted as shown in Table 1. It is worth noting that melt viscosity will rise with increased CNTs content and even changes from a viscous fluid to an elastic solid with high CNTs content [16]. In this research, some samples were molded at a higher melt temperature to improve liquidity. However, if the CNTs content was 9 wt.%, even with higher temperatures, the melt viscosity was still too high to form a secondary flow, so we cannot get the IntM sample with 9 wt.% of CNTs.

CM: Provides a low shear condition (Figure 1a). Plates were compression molded with 1.2 mm thickness at 200 °C. Subsequently, the plate was cut into a dumbbell shape.

IM: Provides a moderate shear condition (Figure 1b). Melt filled the mold chamber under constant pressure. Dumbbell-type samples with 2 mm were prepared using a micro conventional injection molding (SJZS-10A, Wuhan Ruiming Co., Ltd., Wuhan, China) with a mold temperature of 50 °C. A barrel temperature of 200 °C was suitable for most samples, except the 9IM, which needed 210 °C for complete filling due to the high melt viscosity.

IntM: Provides a high shear condition (Figure 1c). The melt filled the mold in two flows. A gradient pressure setting achieved this: The initial injection pressure was too low to fill the cavity. After 1 s of this short shot, the filling was completed by a higher injection pressure. It is worth mentioning that most commercial injection equipment can implement this pressure setting to magnify the shear stress. In this manuscript, IntM samples were also prepared using the micro conventional injection molding (similar to IM). Except for 6IntM, prepared under 210 °C barrel temperature, other samples were molded at 200 °C.

### 2.2. Sample Testing

#### 2.2.1. Measurement of Mechanical Properties

The tensile test was conducted at room temperature (26 °C) on an electro-universal testing machine (GOTECH-20KN, GOTECH Testing Machines Co., Ltd., Dongguan, China) with a 50 mm/min cross-head speed. All tests were conducted along the injection direction, and more than five sample properties were calculated.

#### 2.2.2. Measurement of Electrical Conductivity

Electrical conductivity was performed using the Keithley 6487 source meter (Tektronix Inc., Shanghai, China). Both ends of the specimens were coated with conductive silver paint. The electrical conductivity (σ) was then calculated according to the following:σ = L/RA(1)
L is the distance between the electrodes, R is the measured resistance, and A is the cross-sectional area. For IM and IntM samples, the σ was measured along the flow and transverse directions. The values of σ were calculated as the average of four samples.

#### 2.2.3. X-ray Measurements

The synchrotron 2D-WAXD experiment was carried out on a HomeLab (Rigaku, Japan) to calculate the molecular chain orientation (M_ori_) of a sample. The dimension of the rectangle-shaped beam was 100 × 100 μm^2^, the light wavelength was 0.154 nm, and the X-rays were imaged after penetrating the sample.

The crystallinity (*X_c_*) was calculated according to the following [22]:(2)$$X_{c}=\frac{A_{c}}{A_{c}+A_{a}}\times 100\%$$
where *A_c_* and *A_a_* are the fitting intensity of the crystallization and amorphous peaks. The orientation of lamellar crystals in the injection-molded parts was calculated using the Hermans orientation function. In this method, the crystal orientation was characterized by the average orientation of the normal to the crystal plane concerning an external reference frame. Accordingly, the flow direction was taken as the reference direction. For a set of hkl planes, the average orientation, which can be expressed as $\left({cos}^{2}\varphi \right)$_hkl_, was calculated mathematically using the following equation [23]:(3)$${\left({cos}^{2}\varphi \right)}_{hkl}=\frac{\int_{0}^{\pi /2}I(\varphi ){cos}^{2}\varphi sin\varphi d\varphi }{\int_{0}^{\pi /2}I(\varphi )sin\varphi d\varphi }$$
with φ being the azimuthal angle and I(*φ*) being the scattered intensity along the angle φ. Herman’s orientation function, f, was defined as
(4)$$f=\frac{{3({cos}^{2}\varphi )}_{hkl}-1}{2}$$
with f having a value of −0.5 with the normal of the reflection plane being perpendicular to the reference direction (*φ* = 90°), a value of 1 with the normal of the refection plane parallel being the reference direction (*φ* = 0°), and a value of 0 with the orientation being random.

#### 2.2.4. Optical Microscopy (OM)

Thin slices cut using microtome were used for optical microscopy observations. The observation zones were along the flow direction (FD) or transverse direction (TD) as shown in Figure 2d. The slice thickness was fixed at 5 μm, and ImageJ was used to process the images after removing the non-clear parts from these images. More than nine microscopy photos for each sample were used to calculate the A_dis_ (CNTs agglomerate dispersion) and A_ori_ (CNTs agglomerate orientation). Moreover, using OM to count too-small agglomerates can easily lead to errors, so only agglomerates with an area greater than 10 μm^2^ were measured.

#### 2.2.5. Scanning Electron Microscope (SEM)

A scanning electron microscope (ZEISS Gemini 300, ZEISS Microscopy LLC, Oberkochen, Germany) was used for fracture surface observations. The fracture surface was gold sputtered after being yield stretched.

## 3. Results and Discussion

### 3.1. Sample Structure

#### 3.1.1. CNTs Orientation

As shown in Figure 2d, IM and IntM samples were molded into dumbbell-type shapes, and the OM images of 2IM were obtained from both FD and TD. Oriented CNTs agglomerates can be observed in FD, as shown in Figure 2a, a typical three-layer structure is formed. As the name implies, the frozen layer is in contact with the cold mold wall, where polymer melts can be “frozen” instantly and unable to form oriented CNTs agglomerates. Shear layers, as shown in Figure 2a,b, are where a large number of CNTs agglomerates were oriented along the flow direction, while in the core layer, the CNTs agglomerates returned to a spherical form. In contrast, no oriented structure can be found in TD, as shown in Figure 2e,f. It means all CNT agglomerates in 2IM were oriented along FD and perpendicular to TD. So, this research used images captured from FD to calculate agglomerate orientation (A_ori_), while figures from TD were used to calculate agglomerate dispersion (A_dis_), in order to avoid the influence of orientation. The specific calculation method will be introduced later.

OM images of 2CM, 2IM, and 2IntM in FD are shown in Figure 3. Shear can change the morphologies of CNTs agglomerates; for example, no oriented structure can be found in 2CM, while both IM and IntM can align CNTs agglomerates. The shear layer thickness of 2IntM was much greater than that of 2IM as shown in Figure 3b,c. As the enlarged images (Figure 3e,f) show some oriented CNTs agglomerates were longer than 100 μm along the flow direction. They were expected to form a conductive path in the flow direction with increased CNTs content, and their effects on mechanical properties will be discussed later.

The length-diameter ratio of CNTs agglomerate was calculated from OM images for different CNT content and processing methods. If CNTs agglomerate is abstracted into a cylinder, the greater its orientation, the larger the length-diameter ratio will be. So, as shown in Figure 4a, the length-diameter ratio of CNTs agglomerate was calculated and named as A_ori_. With the help of Image J, more than six OM pictures were used for each sample to calculate A_ori_.
A_ori_ = L_long_/L_short_(5)
where L_long_ is the length of the CNTs agglomerate in the longest direction, and L_short_ is the length of the CNTs agglomerate in the shortest path. Larger A_ori_ value suggests high agglomerate CNTs orientation.

Because both oriented and non-oriented CNT agglomerates existed in each sample, the A_ori_ variance is significant. As displayed in Figure 4b, the A_ori_ of IntM samples was always larger than that of IM and CM because the secondary flow formed a wider shear layer. In addition, CM-A_ori_ did not equal 1 but remained about 2, because, before compression molding, the CNTs agglomerate had been stretched in the extrusion colling process. This stretched oriented structure can be retained at 200 °C. It is worth noticing that these oriented structures have been cut short and rearranged in CM, so they will not show anisotropy on the macro level. In general, although very low shear existed in the compression molding process, randomly orientated agglomerates remained. It is worth noting that when CNT content is higher than 6 wt.%, some CNTs agglomerates will connect and form a continuous phase (Figure 4d), which can hinder statistics on the shape of individual agglomerates, and A_ori_ loses its value. In this way, if CNTs content was higher than 4 wt.%, A_ori_ could not be used to describe the orientation.

Compared with A_ori_, every sample’s molecular chain orientation (M_ori_) can be explored through the transmission mode of WAXD. Because CNTs flowed with PP during the molding process, M_ori_ can reflect the CNTs orientation, especially for these dispersed CNTs. WAXD images are shown in Figure 5. For CM and IntM samples, such as 2CM, 6CM, 0IntM, 2IntM, and 6IntM, the orientation almost did not change with CNTs content, so only parts of them were selected for WAXD testing. The diffraction ring of CM always presented a complete circle, representing its random orientation structure. In contrast, the IntM sample presented a prominent arc bright spot inside the ring, indicating that it exists in both high orientation and random structure. Correspondingly, IM samples showed some IntM-like diffraction rings, but the orientation rises with increased CNTs content, which will be discussed later.

The (040) α plane was calculated to estimate M_ori_. Figure 6a shows the azimuth angle of the CM, IM, and IntM samples with 2 or 6 wt.% CNTs. CM samples had a very gentle curve, while the peaks, denoted with arrows, indicated high orientation degrees formed using IM and IntM. In Figure 6b, the M_ori_ can be separated into HIGH, MEDIUM, and LOW areas. The dividing standards are shown in Table 2. When the CNTs content was lower than 6 wt.%, M_ori_ showed a similar trend to A_ori_ as shown in Figure 3: CM, IM and IntM increased in turn.

For CM samples, the M_ori_ was lower than 0.5, and all were located in the LOW area. In comparison, all the IntM samples M_ori_ were higher than 0.9 and located in the HIGH area. However, for IM samples, the orientation changed with CNTs content. When the CNT content was low, its M_ori_ was located between CM and IntM, which was situated in the MEDIUM region. However, M_ori_ jumped to the HIGH part after the CNT content reached 6 wt.%. Increased M_ori_ attributes to added rigid particles such as CNTs, improve both the storage and the loss modulus, and can help increase the shear area [21,24]. In addition, CNTs not only improved the melt viscosity but also acted as a nucleation site to freeze the oriented molecular chains and form a hybrid shish-kebab structure, which can super improve the mechanical performance [25,26,27]. That is to say, with increased CNTs content, IM can obtain a high shear stress similar to IntM and make M_ori_ greater than 0.91. It is worth mentioning that the 9IM-M_ori_ p slightly because the molding temperature was 10℃ higher than others. The higher melt temperature made it easier for the oriented molecular chains to recover, and in higher CNTs content, the entangled CNTs network also hindered the orientation. Even though the 9IM-M_ori_ was lower than 6IM-M_ori_, the 9IM-M_ori_ was still located in the high area.

#### 3.1.2. CNTs Dispersion

In order to quantity dispersion degree, we introduce the parameter of CNTs aggerate dispersion (A_dis_). The number of CNTs aggregates per unit area is used to represent the dispersion. OM Images obtained from TD were used to calculate A_dis_. A_dis_ is equal to the number of agglomerates within 1 mm^2^, and the A_dis_ can be obtained using Equation (6):(6)Adis=106×AnumberAarea
where A_number_ represents the number of agglomerates, A_area_ is the OM images area in μm^2^. When an aggregate has been completely aggregated into one mass, that is, the dispersion is the worst, A_dis_ is equal to 1. As the aggregate disperses into particles, A_dis_ will gradually increase. The results of A_dis_ are shown in Figure 7. When the CNTs content was 1 wt.%, the A_dis_ of 1IntM was significantly higher than that of 1CM and 1IM, indicating that the shear formed by secondary flow can further promote the dispersion of CNTs. The large-size CNTs agglomerates can be ruptured by the high shear force, doubling the number of agglomerates per unit area. Other researchers have also reported that CNTs achieved a relatively better distribution when processed under high-shear conditions [16,28]. When CNTs content increased to 2 wt.%, such as 2IntM, about 6000 CNTs agglomerates existing in 1 mm^2^, while for CM samples, the number was relatively small, such as for 2CM which achieved 4000 CNTs agglomerates. It means that the diameter of the CNT agglomerates in CM was larger than IM and IntM, or some adjacent CNTs agglomerates have been interconnected and integrated. The A_dis_ of each sample will reach the maximum value if the CNTs content increases to 4 wt.%.

If the CNTs content is increased to 6 wt.%, the A_dis_ of 6IM and 6CM drops to nearly 1200, indicating that the CNTs agglomerates cannot maintain a high dispersion at high CNTs content but combine to form larger agglomerates. Even if the secondary shearing of IntM can promote the distribution of CNTs, the A_dis_ of 6IntM drops to around 3000. When the CNTs content reached 9 wt.%, the A_dis_ of 9CM reduced to 20, and most CNTs were joined together. Meanwhile, although the A_dis_ of 9IM was slightly higher by shearing, the number of agglomerates in 1mm^2^ also decreased to 400. The decreased A_dis_ and the formation of a continuous CNTs phase was expected to improve the conductivity, tensile strength, and modulus, which will be discussed in the next section.

### 3.2. Effects of Orientation and Dispersion

#### 3.2.1. Orientation Effect on Yield Properties

Figure 8 shows the relationship between tensile properties and the CNTs content of the samples obtained by different molding methods. These colors are used to divide the three M_ori_ types, which are high, middle, and low. The dividing standards are already shown in Table 2, and the selectivity yield curves are given in support information in Appendix A. Regardless of the content of CNTs, CM, and IntM samples were located in the LOW and HIGH ranges, respectively. In contrast, IM-M_ori_ can leap from Medium to High ranges when CNTs content reaches 6 wt.%. Due to the formation of oriented structures in PP/CNTs nanocomposites under high shear conditions, such as shish-kebab, hybrid shish-kebab, and oriented CNTs, the tensile strength and toughness of the material can be improved. Under the same CNT content, the tensile strength of the IntM sample was always higher than that of IM and CM due to the higher M_ori_ (Figure 8a). The tensile strength of the IntM and CM samples did not change significantly, but the tensile strength of IM samples greatly increased when M_ori_ jumped to HIGH ranges when the CNTs content reached 6 wt.%. This is also due to the massive formation of orientation structure at high M_ori_. As shown in Figure 8b, with the increased CNTs content, the tensile modulus of all samples showed an increasing trend. It has been proved that CNTs can improve the strength and stiffness of PP at the expense of toughness, because CNTs can induce α-nuclei in PP and increase the crystallinity leading to superior stiffness [29,30,31]. On the other hand, shear can promote the formation of oriented crystals, further improving the modulus. For example, the modulus of 6IntM was higher than that of 6IM and 6CM. In general, within the higher CNTs content, higher M_ori_ always brought a higher strength and modulus [32]. As a result, the tensile strength of 4IntM was 13% higher than that of 0CM, while the tensile modulus of 5IntM was 35% higher than that of 0CM.

Figure 9a shows the Low M_ori_-CNTs agglomerates in the tensile fraction surface, which is smooth and shows many CNTs agglomerates. In the fracture mode, as shown in Figure 9a, the CNTs agglomeration acts as stress concentration points, which is unfavorable for enhancing mechanical properties. Agglomeration can induce the generation of cracks, and the cracks are easy to spread along the interface of the un-oriented agglomerates without damaging CNTs. So, Low M_ori_-CNTs agglomerates are hard to exert a high mechanical strength [33]. However, cracks cannot bypass the CNTs agglomerates for high M_ori_ products but destroy them. This is due to the CNTs bridge effect which has been highlighted for toughening mechanisms, since CNTs can induce crack propagation and crack deviation to increase fracture toughness [34]. In the destruction process, some CNTs were pulled out of the matrix, and good wettability allowed the PP matrix to be coated on CNTs, thus forming many fibers on the fracture surface, as shown in Figure 9b. This process can prevent the rapid propagation of cracks and improve the tensile strength of the parts.

#### 3.2.2. Orientation Effect on Electrical Conductivity

The electrical conductivity (σ) of various samples were measured in FD and TD directions. As shown in Figure 10a, samples with different M_ori_ values showed different percolation thresholds. That is, the sample conductivity underwent multiple orders of magnitude transitions at a certain CNTs content. For CM samples with low M_ori_, the percolation threshold was 4 wt.%, while for IM and IntM samples with high M_ori_, the percolation threshold was 6 wt.% in FD and 9 wt.% in TD, respectively. This phenomenon was also found by other researchers, such as Ameli et al. [35] who studied injection molded PP/CNTs nanocomposite foams, and their results showed low conductivity which was attributed to the one-dimensional oriented CNTs in the flow direction. This is consistent with our previous research conclusions on PP/PA/CNTs blends. When CNTs choose to disperse in the PA phase, although the elongated PA phase holds advantages in terms of its tensile properties, its conductivity network is defective [36]. In general, orientated conductive phases, such as elongated CNTs agglomerates or PA/CNTs phases, are relatively hard to form a conductive path.

In addition, a higher orientation brings anisotropy while, at the same time, reducing conductivity. Figure 10b shows the samples anisotropy intensity derived using σ_FD_/σ_TD_. σ_FD_ and σ_TD_ are the conductivity in FD and TD directions, separately. For 6IM, 6IntM, and 9IM samples, the conductivity in FD was about four or five orders of magnitude higher than that in TD. These samples can be called anisotropic polymer composites, which have recently attracted much attention for their essential applications [37]. In general, a lower orientation helps to improve conductivity and reduce the percolation threshold, while a higher orientation leads to anisotropy.

#### 3.2.3. Combined Influence by Orientation and Dispersion

To clarify the combined influence of orientation and dispersion, the area of circles has been used to express the electrical conductivity, as shown in Figure 11a. M_ori_ is divided into three levels (LOW, MEDIUM, and HIGH), differentiated using color. When the M_ori_ is LOW, higher conductivity can be obtained even if A_dis_ is high; however, for other samples, the lower the A_dis_ value is, the higher conductivity will be. The lower A_dis_ helps oriented CNTs form a conductive network, while un-oriented CNTs can form a conductivity path regardless of A_dis_.

On the other hand, as shown in Figure 11b, the area of circles can be used to express the yield strength. The tensile strength is highly correlated with M_ori_ and almost unaffected by A_dis_, meaning high tensile strength can only be obtained in the high M_ori_ region. Meanwhile, the tensile strength shows no change with the increased dispersion degree.

The conductivity of the lower M_ori_ sample is schematically shown in Figure 12a. The agglomerates and dispersive CNTs remained randomly arranged. The electronics can be transferred along the horizontal and vertical directions, which has a high probability of forming a conductive pathway in three-dimensions. Even if the agglomerates do not connect, they can still develop conductive pathways via a physical connection among individual fillers. So, the low-oriented CM sample obtained the lowest percolation threshold at 4 wt.%. However, when M_ori_ was high, as shown in Figure 12b, with the orientated CNTs and CNTs agglomerates, electrons lost their ability to propagate in the vertical direction. The CNTs content must be increased to form a pathway in FD. Correspondingly, in FD, the percolation threshold of highly oriented IM and IntM samples rises to 6 wt.%. In high orientation, the conductive path in TD is the most difficult to form and has the lowest conductivity. In addition, shear provided by the injection molding can improve the CNTs interface and even generate a hybrid shish-kebab. Hence, a layer of PP matrix surrounding CNTs impairs the enhancement of σ [38,39]. Ultimately, the conductive path in TD is formed until A_dis_ is less than 100, which corresponds to the highest percolation threshold of 9 wt.%.

M_ori_ can constrain the effect of A_dis_ on conductivity. For low-oriented products (Figure 12c), the A_dis_ does not affect the conductivity because even at high A_dis_, there is an opportunity to construct a three-dimensional conductive pathway for randomly distributed CNTs agglomerates. However, for oriented CNTs agglomerates (Figure 12d), high A_dis_ is more likely to destroy their conductive pathways in the flow direction, thus increasing the percolation threshold.

## 4. Conclusions

CM, IM, and IntM prepared PP/CNTs composites, which could provide different shear strengths. The dispersion and orientation of CNTs were quantified using A_ori_ (Agglomerate orientation), M_ori_ (Molecular orientation), and A_dis_ (Agglomerate orientation) at both aggregate and molecular chain scales. The results showed that high shear can increase the A_ori_, M_ori_ and A_dis_.

Increasing both orientation and CNTs content can increase the tensile strength and modulus by 13% and 35%, respectively, but dispersion degree at the micro level has little effect on the tensile strength. The oriented CNTs can form many fibers at the fracture interface and inhibit crack propagation. Thus, a high orientation results in a high tensile strength and modulus. Increasing CNTs content, the M_ori_ of CM and IntM almost did not change, but M_ori_ of the IM samples realized a transition from medium to high. Meanwhile, the tensile strength and modulus of IM also jumped with M_ori_. On the other hand, A_dis_ almost does not affect the tensile properties.

Both high orientation and high dispersion are averse to the formation of conductive pathways. Three percolation thresholds of 4, 6, and 9 wt.% can be obtained under three special orientations and dispersion states of CNTs, respectively. The conductivity is affected by both CNTs orientation and dispersion. The conductivity percolation threshold of low M_ori_ samples is 4 wt.%, the CNTs are more likely to form a three-dimensional conductive pathway, and A_dis_ do not affect the conductivity. However, the percolation threshold of high M_ori_ samples in FD is raised to 6 wt.%, and in TD is increased to 9 wt.%. High M_ori_ samples get an anisotropy of nearly six orders of magnitude in FD and TD, which can already be used as anisotropically conductive polymer composites. High M_ori_ samples cannot form a three-dimensional pathway but prioritize creating a pathway in the FD, then forming a pathway in TD with further increased CNTs content. Meanwhile, the value of A_dis_ is negatively correlated with the conductivity; that is, high conductivity can only be obtained under low CNTs aggregation dispersion.

In general, both high orientation and high dispersion of CNTs agglomerate against the formation of conductive network, while a high orientation can improve the mechanical properties.

## Figures and Tables

**Figure 1 polymers-15-02370-f001:**
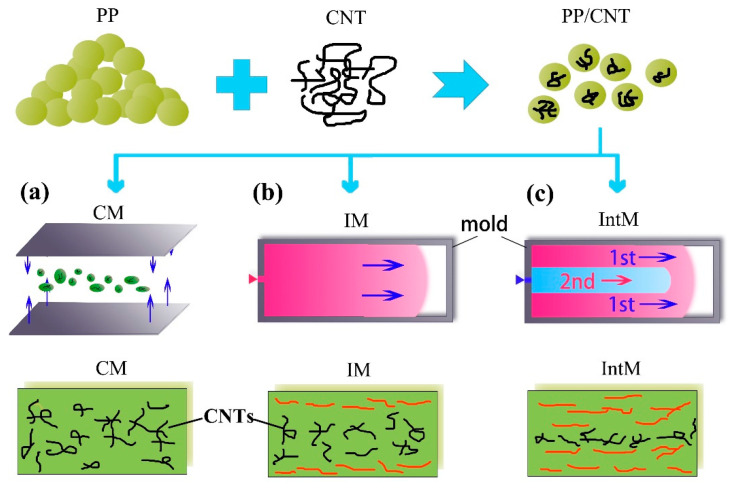
Schematic of molding methods and CNTs orientation. (**a**) Compression molding (CM), (**b**) conventional injection molding (IM), (**c**) intermittent injection molding (IntM).

**Figure 2 polymers-15-02370-f002:**
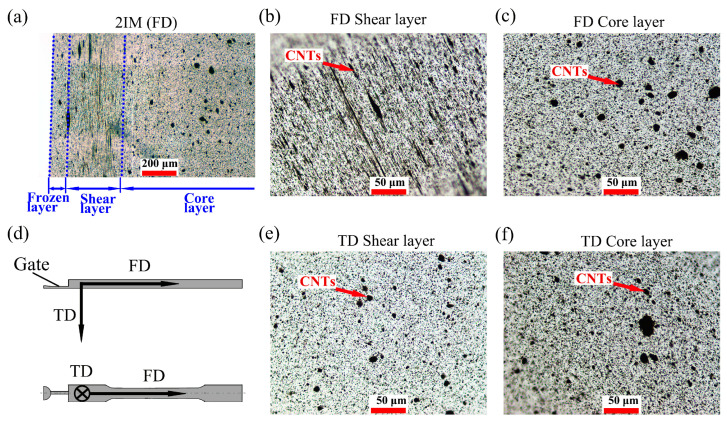
OM picture of 2IM, (**a**) overall image of 2IM in FD, (**b**) enlarged detail of shear layer in FD, (**c**) enlarged detail of core layer in FD, (**d**) schematical of samples, (**e**) enlarged detail of shear layer in TD, (**f**) enlarged detail of core layer in TD.

**Figure 3 polymers-15-02370-f003:**
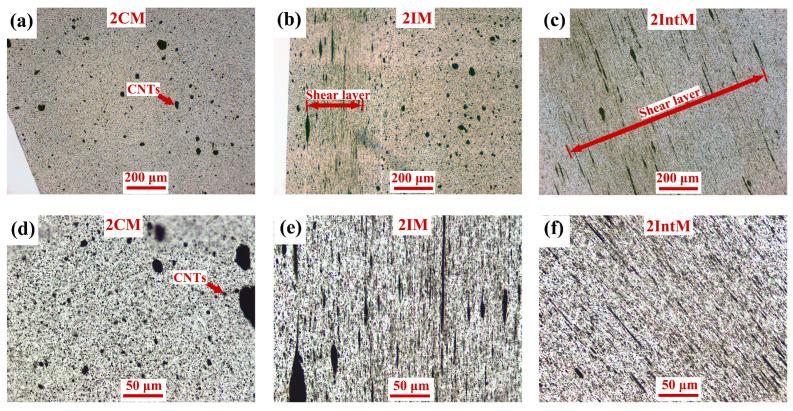
OM images in FD, (**a**,**d**) 2CM, (**b**,**e**) 2IM, (**c**,**f**) 2IntM.

**Figure 4 polymers-15-02370-f004:**
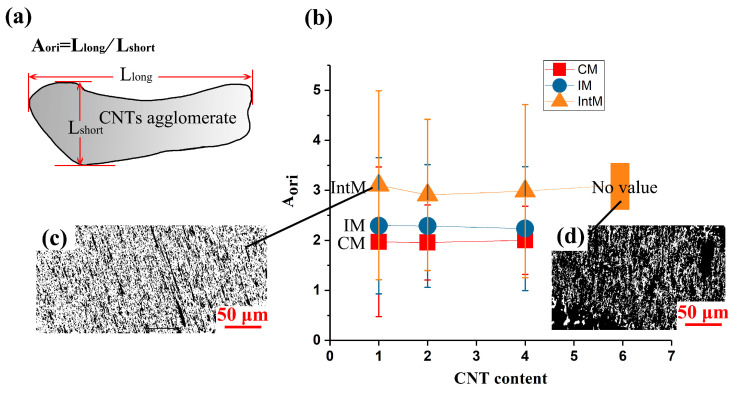
(**a**) Calculation method of Agglomerate orientation degree (A_ori_), (**b**) A_ori_ of CM, IM, and IntM change with CNT content, (**c**) OM image of 1IntM, (**d**) OM image of 6IntM.

**Figure 5 polymers-15-02370-f005:**
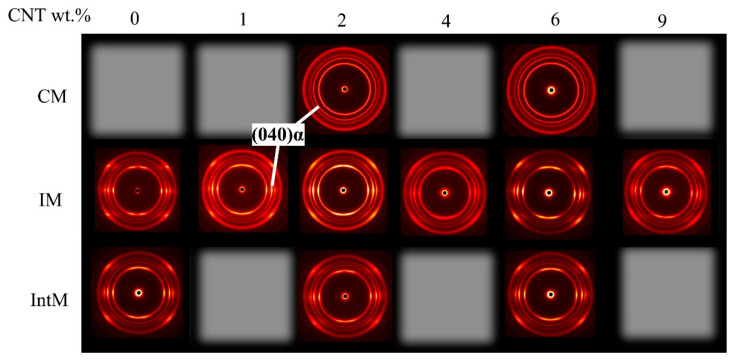
2D-WAXD patterns of samples with different molding methods and CNTs content.

**Figure 6 polymers-15-02370-f006:**
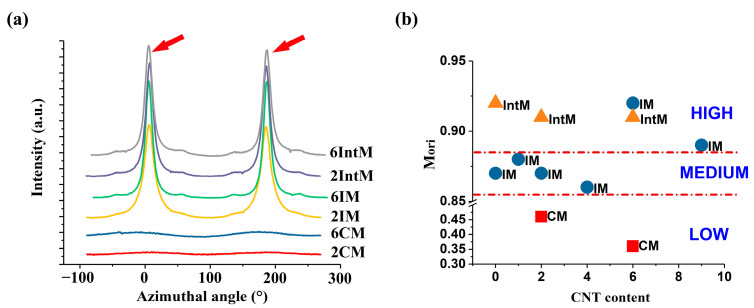
(**a**) Azimuthal profiles taken at the (040) α reflection as a function of azimuthal angle, (**b**) Degree of molecular orientation (M_ori_) of CM, IM, and IntM calculated from Azimuthal profiles.

**Figure 7 polymers-15-02370-f007:**
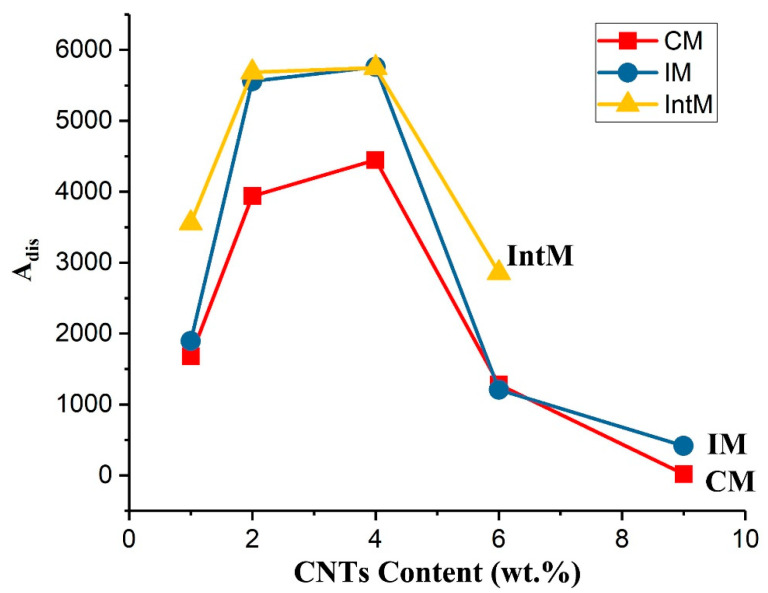
Agglomerate dispersion degree (A_dis_) as a function of CNTs content.

**Figure 8 polymers-15-02370-f008:**
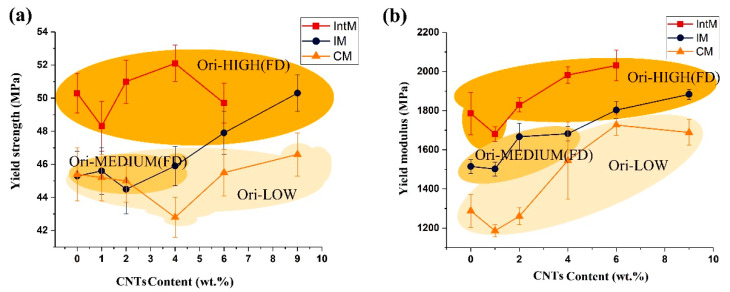
(**a**) Yield strength, (**b**) yield modulus along FD as a function of CNTs content.

**Figure 9 polymers-15-02370-f009:**
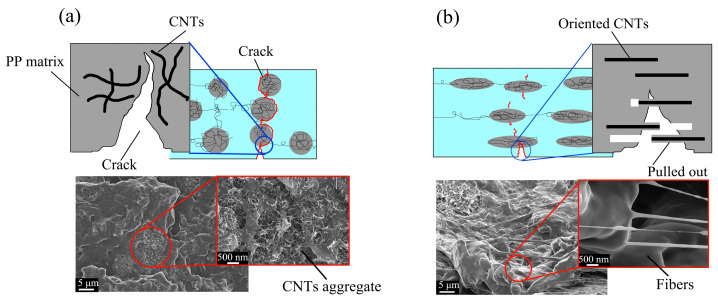
Yield schematic and SEM images of fracture surface after yield. (**a**) Low M_ori_ CNTs agglomerates, (**b**) High M_ori_ CNTs agglomerates.

**Figure 10 polymers-15-02370-f010:**
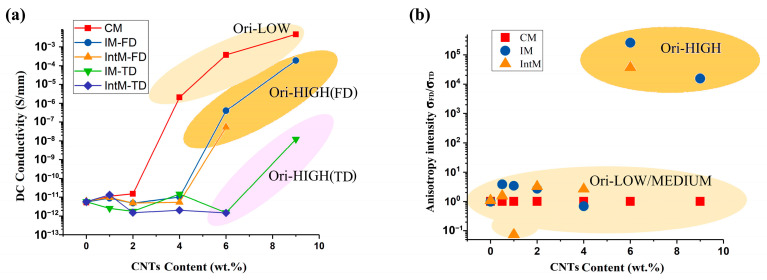
(**a**) Direct current conductivity of various samples as a function of CNTs content, (**b**) anisotropy intensity σ_FD_/σ_TD_.

**Figure 11 polymers-15-02370-f011:**
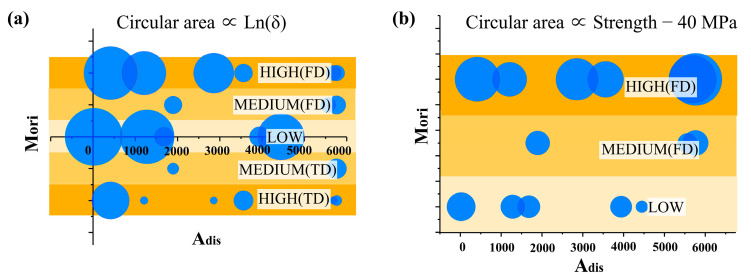
Agglomerate dispersion degree (A_dis_) versus Molecular orientation (M_ori_). (**a**) the electrical conductivity and (**b**) the yield strength of the samples is expressed using the area of circles.

**Figure 12 polymers-15-02370-f012:**
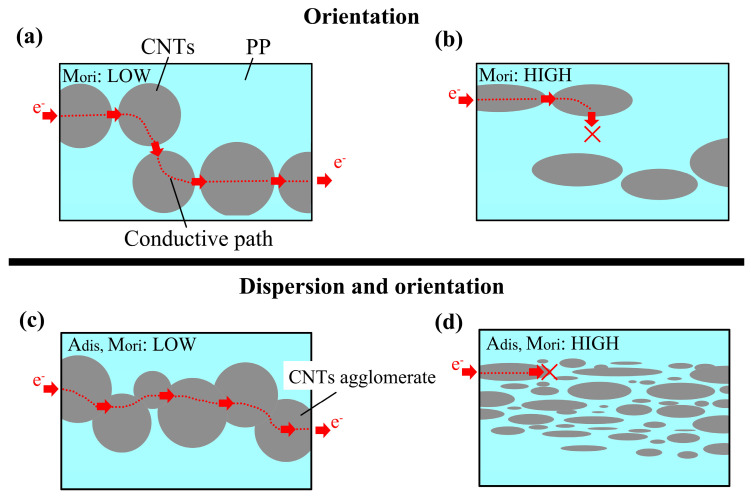
Schematic of the conductivity for (**a**) Low M_ori_, (**b**) High M_ori_, (**c**) Low A_dis_ and M_ori_, (**d**) High A_dis_ and M_ori_ samples.

**Table 1 polymers-15-02370-t001:** Molding parameters of three molding methods.

	1CM-6CM	1IM-5IM	6IM	1IntM-4IntM	5IntM
Melting temperature (℃)	200	200	210	200	210
CNTs content (wt.%)	1–6	1–5	6	1–4	5
First Injection pressure (bar) *		5.5	5.5	2	2
Second Injection pressure (bar) *				5.5	5.5

* Note: Injection pressure is the set pressure of the equipment, not the melt pressure.

**Table 2 polymers-15-02370-t002:** Criteria for the level of orientation.

	CM (0~9 wt.%CNT)	IM (0~4 wt.%CNT)	IM (6~9 wt.%CNT)	IntM (0~6 wt.%CNT)
Mori	0~0.5	0.8~0.88	0.91~0.92	0.91~0.92
level-FD	LOW	MEDIUM (FD)	HIGH (FD)	HIGH (FD)
level-TD	LOW	MEDIUM (TD)	HIGH (TD)	HIGH (TD)

## Data Availability

Some or all data, models, or code generated or used during the study are available from the corresponding author by request.

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
