# Peer review of "Effects of Orientation and Dispersion on Electrical Conductivity and Mechanical Properties of Carbon Nanotube/Polypropylene Composite"

_polymers, 2023, doi:10.3390/polym15102370_

Round 1
Reviewer 1 Report
Dashan Mi et al. reported Effects of orientation and dispersion on electrical conductivity and mechanical properties of carbon nanotube/polypropylene composite. This manuscript needs major revision before publication. The major problems are as follows:
1. Please highlight the scientific novelty of this study.
2. The quality of Figure 2 needs improvement.
3. Please indicate the dispersion of the CNT in the PP matrix phase using an arrow mark.
4. To enhance the quality of the manuscript, it is recommended to include DMA, tensile and flexural properties of the samples.
5. Please ensure that the reference format follows the guidelines provided by the polymers journal.
6. The introduction section should provide more information and include relevant articles on CNT-loaded composites to give readers a comprehensive background.
a. Enhanced thermal stability, toughness, and electrical conductivity of carbon nanotube-reinforced biodegradable poly(lactic acid)/poly(ethylene oxide) blend-based nanocomposites.
b. Evident improvement in burning anti-dripping performance, ductility and electrical conductivity of PLA/PVDF/PMMA ternary blend-based nanocomposites with additions of carbon nanotubes and organoclay
c. Polyamide 6/Poly(vinylidene fluoride) Blend-Based Nanocomposites with Enhanced Rigidity: Selective Localization of Carbon Nanotube and Organoclay.

Minor editing of English language required.
Author Response
Dear Reviewer:
Thank you for your comments. We have studied the comments carefully and have made corrections which we hope meet with approval.
Q: 1. Please highlight the scientific novelty of this study.
A: Thanks for your advice; some relevant content has been added.
Introduction: “This paper proves that the high dispersion of CNTs agglomerate goes against forming a conductivity network. At the same time, the increased orientation of CNTs causes the electric current to flow only in the orientation direction.”
Results and discussion: “In general, within the higher CNTs content, higher Mori always brings a higher strength and modulus. As a result, the tensile strength of 4IntM is 13% higher than that of 0CM, while the tensile modulus of 5IntM is 35% higher than that of 0CM.”
Conclusion: “Increasing both orientation and CNTs content can increase the tensile strength and modulus by 13% and 35%, respectively, but dispersion degree at the micro level has little effect on the tensile strength.” “Both high orientation and high dispersion are averse to the formation of conductive pathways. Three percolation thresholds of 4, 6, and 9 wt.% can be obtained under three special orientations and dispersion states of CNTs, respectively.”
Q: 2. The quality of Figure 2 needs improvement.
A: Thanks for your comment. Figure 2 has been reformatted to increase the size, improve contrast, and make the text clearer.
Q: 3. Please indicate the dispersion of the nano-fillers in the matrix phase using an arrow mark.
A: Thanks for your advice; arrows are added in Figures 2 and 3 to indicate the dispersion of CNTs.
Q: 4. To enhance the quality of the manuscript, it is recommended to include DMA tensile and flexural properties of the samples.
A: Thanks for your recommendation. The DMA can help to give more information about mechanical properties. However, the above experiments cannot be completed now due to insufficient relevant conditions. The influence of dispersion and orientation on more mechanical properties is expected to be further discussed in subsequent research.
Q: 5. Please ensure that the reference format follows the guidelines provided by the polymers journal.
A: Thanks for your advice. The reference format is modified to follow the guidelines.
Q: 6. The introduction section should provide more information and include relevant articles on CNT-loaded composites to give readers a comprehensive background.
A: Thanks for your advice. These relevant articles can help to give a more comprehensive background. The manuscript has been revised accordingly.
“Using carbon nanotubes (CNTs) as nanofillers has increased in applications 4-7, such as burning anti-dripping performance, electromagnetic shielding, and sensing element. Polymer blend-based multicomponent nanocomposites with superior mechanical and thermal properties due to the modification of phase morphology were reported 8.”
“However, cracks cannot bypass the CNTs agglomerates for high Mori products but destroy them. This is due to the CNTs bridge effect which has been highlighted for toughening mechanisms since CNTs can induce crack propagation and crack deviation to increase fracture toughness 34.”
References:
- Deng, X.; Xie, S.; Wang, W.; Luo, C.; Luo, F., From carbon nanotubes to ultra-sensitive, extremely-stretchable and self-healable hydrogels. European Polymer Journal 2022, 178, 111485.
- Zhang, R.; Yan, W.; Yang, Q.; Chen, X.; Chen, K.; Ding, Y.; Xue, P., Analysis of thermal‐active bending and cyclic tensile shape memory mechanism of UHMWPE/CNT composite. Polymer Composites 2022.
- Zhang, Z.; Bellisario, D.; Quadrini, F.; Jestin, S.; Ravanelli, F.; Castello, M.; Li, X.; Dong, H. J. P., Nanoindentation of multifunctional smart composites. 2022, 14 (14), 2945.
- Behera, K.; Chen, J.-F.; Yang, J.-M.; Chang, Y.-H.; Chiu, F.-C., Evident improvement in burning anti-dripping performance, ductility and electrical conductivity of PLA/PVDF/PMMA ternary blend-based nanocomposites with additions of carbon nanotubes and organoclay. Composites Part B: Engineering 2023, 248, 110371.
- Lin, H.-M.; Behera, K.; Yadav, M.; Chiu, F.-C., Polyamide 6/poly (vinylidene fluoride) blend-based nanocomposites with enhanced rigidity: Selective localization of carbon nanotube and organoclay. Polymers 2020, 12 (1), 184.
- Behera, K.; Chang, Y.-H.; Yadav, M.; Chiu, F.-C., Enhanced thermal stability, toughness, and electrical conductivity of carbon nanotube-reinforced biodegradable poly (lactic acid)/poly (ethylene oxide) blend-based nanocomposites. Polymer 2020, 186, 122002.

Reviewer 2 Report
Dear,
The authors investigated the effect of orientation and dispersion on the electrical conductivity and mechanical properties of the polypropylene/carbon nanotube composite. Therefore, the manuscript has merit for publication. In addition, overall, the manuscript is well-written and reasoned. Small details:
> Please report whether the results obtained from the composites can be applied to astatic packaging, electromagnetic shielding, etc;
> How does the molding method affect the relationship between the intensity of the D band and the G band? Did the authors not perform Raman spectroscopy?
> Why did the authors apply a 50 mm/min speed in the tensile test for composites? Generally, the test is conducted at low speed;
Minor editing of English language required.
Author Response
Dear Reviewer:
Thank you very much for your comments. We have studied the comments carefully and have made corrections which we hope meet with approval.
Q: Please report whether the results obtained from the composites can be applied to astatic packaging, electromagnetic shielding, etc;
A: Thanks for your advice. These results may not be applied to astatic packaging but can help to produce shielding and computer chip. The relevant contents are modified.
“In the present work, PP was used as the matrix to prepare PP/CNTs nanocomposites, which has potential applications in the computer chip, electromagnetic shielding, and dissipation of electrostatic discharge 1.” “Then the dispersion and orientation of CNTs are quantified and used to study the relationship with performance, which can help to prepare PP/CNTs nanocomposites on demanded mechanical and electrical properties.
”
Q: How does the molding method affect the relationship between the intensity of the D band and the G band? Did the authors not perform Raman spectroscopy?
A: Thanks for your comments. The molding method can affect the D and G bands. However, both Raman spectroscopy and WAXD can give information about orientation degrees in different scales. This paper used POM and WAXD to measure the orientation of aggregates and molecular chains, and the orientation at different scales was described. Therefore, the difference between D and G bands was not discussed.
Q: Why did the authors apply a 50 mm/min speed in the tensile test for composites? Generally, the test is conducted at low speed;
A: Thanks for your question. For PP blends, 50 mm/min is a standard tensile test rate2-4. This speed may facilitate the comparison between documents. In addition, CNTs in the conducting polymer act as a stress concentration, resulting in short elongation at break, so if the tensile rate is too fast, it will be difficult to reflect the change in elongation at break.
References:
- Gulrez, S. K.; Ali Mohsin, M.; Shaikh, H.; Anis, A.; Pulose, A. M.; Yadav, M. K.; Qua, E. H. P.; Al‐Zahrani, S., A review on electrically conductive polypropylene and polyethylene. Polymer composites 2014, 35 (5), 900-914.
- Huan, Q.; Zhu, S.; Ma, Y.; Zhang, J.; Zhang, S.; Feng, X.; Han, K.; Yu, M., Markedly improving mechanical properties for isotactic polypropylene with large-size spherulites by pressure-induced flow processing. Polymer 2013, 54 (3), 1177-1183.
- Du, H.; Zhang, Y.; Liu, H.; Liu, K.; Jin, M.; Li, X.; Zhang, J., Influence of phase morphology and crystalline structure on the toughness of rubber-toughened isotatic polypropylene blends. Polymer 2014, 55 (19), 5001-5012.
- Zhang, Z.; Lei, J.; Chen, Y.; Chen, J.; Ji, X.; Tang, J.; Li, Z.-M., Tailored structure and properties of injection-molded atactic polypropylene/isotactic polypropylene blend. ACS Sustainable Chemistry Engineering 2013, 1 (8), 937-949.

Round 2
Reviewer 1 Report
The authors have addressed all the major concerns, so I recommended it for publication.